# An improved dataset for predicting mammal infecting viruses from genetic sequence information

Tyler Reddy[1]*, Austin Schneider[2], Aaron R. Hall[1], Adam Witmer[1], Nick Hengartner[3]*

**1** CAI-1: Applied Computer Science, Los Alamos National Laboratory, Los Alamos, New Mexico, **2** P-2: Applied And Fundamental Physics, Los Alamos National Laboratory, Los Alamos, New Mexico, **3** T-6: Theoretical Biology and Biophysics, Los Alamos National Laboratory, Los Alamos, New Mexico

* treddy@lanl.gov (TR), nickh@lanl.gov (NH)

## Abstract

There have been several attempts to develop machine learning (ML) models to identify human infecting viruses from their genomic sequences, with varying degrees of success. Direct comparison between models is problematic, because these models are typically trained and evaluated on different datasets with alternative data splitting schemes, features, and model performance metrics. In this paper we present a standardized dataset of mammal infecting and non-infecting viral pathogens, refined from the previous work of Mollentze *et al.* to include the latest literature evidence, roughly doubling the number of curated host-virus records available to the community, and new host target labels, primate and mammal. The new host labels were included for several reasons, including previous reports that classification performance is better at broader taxonomic ranks and the idea that there may be more data for primate infection that might serve as a suitable proxy for zoonotic potential and avoidance of false positives for human infection due to absence of evidence. On this dataset, we report the performance of eight machine learning models for predicting mammal-infecting viruses from their genomic sequences. We find that randomly assigning cases in our improved dataset to training/testing sets, when compared to the original assignments into training/testing in Mollentze *et al.*, increases the overall average ROC AUC of prediction of human infection from **0.663 ± 0.070** to **0.784 ± 0.013**, consistent with the reduction in phylogenetic distance between train and test sets (relative entropy change from 3.00 to 0.08). The broadest host category of mammal infection can be predicted most reliably at **0.850 ± 0.020**. We share our improved dataset and code to enable standardized comparisons of machine learning methods to predict human host infections. Overall, we have presented preliminary evidence that classification of virus host infection is more tractable at higher taxonomic ranks, that unsurprisingly reducing the phylogenetic distance between training and test sets can improve predictive performance, that peptide kmer features appear to be harmful to out of sample model performance, and we are left with the question of whether models for virus

**Data availability statement:** All datasets used in this study are available in our GitHub repository at https://github.com/lanl/ldrd_virus_work. To ensure the long-term stability of our analysis workflow, the underlying viral genomes are stored in the above repository using git lfs, and the hash of the compressed data at the time of publication is 3a3d3f4e81. This helps avoid changes that occur in upstream databases—over time, however, it may be helpful to update the genomes for corrections in future works. At the time of writing, the git lfs bandwidth of the above repository is limited, so we have also provided the dataset on figshare with DOI https://doi.org/10.6084/m9.figshare.30025597. The identities of the viruses in the various training and test sets are available in CSV files at project path viral seq/data, at release version v0.1.0, and at DOI https://doi.org/10.5281/zenodo.17074080. A human readable HTML file that contains the names of viruses, their assigned host labels, and the citation(s) justifying these host labels, has been made available permanently at https://doi.org/10.6084/m9.figshare.31014793, so that the host labels may be inspected and improved over time.

**Funding:** The project was funded by the Laboratory Directed Research and Development (LDRD) grant 20230044DR at Los Alamos National Laboratory, to TR and NH. The funders had no role in study design, data collection and analysis, decision to publish, or preparation of the manuscript.

**Competing interests:** The authors have declared that no competing interests exist.

host prediction can reasonably be expected to perform well in out of sample scenarios given the likelihood that viruses do not share a common ancestor. Consistent with this concern, when the data is resampled such that there is no overlap between viral families in training and test sets (relative entropy $> 24$), models perform no better than random chance at prediction of human infection regardless of whether kmers are included (ROC AUC **0.50 ± 0.08**) or not (ROC AUC **0.50 ± 0.04**).

### Author summary

Determining whether a virus can infect a human or other animal based on its genetic information is useful for assessing the threat level of circulating and newly emerging viruses. Previous studies in this domain have had access to limited datasets, and in this work we nearly double the amount of manually labelled host data for viral infection, so that others may build on it and improve it further. We use machine learning models to rank the likelihood of human and mammal infection for viruses in this improved dataset. Results are consistent with the determination of host infection being more tractable for broader categories of hosts, like mammals, than for specific species, like humans. This may suggest that the prospects are good for improved future models that first screen viruses based on their likelihood of infecting mammals, and then in a second stage for likelihood of human infection. The most challenging scenarios were for predictions of viruses that were not similar to viruses in the training data, and the question remains whether we can expect reasonable generalization of predictive models to completely new viruses given that, at the time of writing, viruses do not appear to share a common ancestor.

## 1 Introduction

Accurate prediction of viral spillover events and emerging threats is crucial to the development of early warning systems to mitigate future pandemics. Such systems are needed because pandemics from spillover events occur regularly. The Spanish flu (1918) – birds, swine flu (H1N1, 2009) – pigs, Zika virus outbreak (2015) – monkeys, and the recent SARS-CoV-2 (2019) – bats, are a few notable examples [1]. Early detection of spillover events is challenging due to the vast diversity of possible zoonotic pathogens that need to be surveilled. The recent efforts to rank pathogens at highest risk of initiating a viral spillover event, including machine-learning models, [2–5] can partially mitigate this challenge. Viral genomic data is generally the most readily available data, especially for newly discovered viruses, whereas other biological descriptors (features), such as protein structures, require a greater investment of time and resources [6,7]. Therefore, machine learning (ML) models trained solely on nucleotide sequences offer a quick and cost-effective method to triage emerging viruses to identify those of greatest concern to human health. Note that training

solely on nucleotide sequences does not mean neglecting structural information, given recent breakthroughs in predicting protein structures from sequence information [8], and with protein language models encoding structural information in embeddings derived from sequence information [9]. Nonetheless, some caution is warranted in structural prediction from sequence information for viruses given that viruses, unlike cellular life, likely do not share a common ancestor [10], and a truly novel pathogenic virus may therefore be less tractable for conventional structure prediction tools.

Many ML classifiers have been developed to identify human infecting viruses from genetic sequence information. They differ significantly with regards to how, and what data, are used, and, how the ensuing results are reported (see Table 1). That heterogeneity and ensuing wide range of ML predictive performances, highlight the need for a benchmark dataset to enable model comparison. Indeed, ML researchers advocate developing ML methods using standardized datasets with agreed upon metrics and reporting standards [11,12].

Several factors should be weighted when designing a standardized reference dataset, given that variation in data selection and reported metrics makes it difficult to compare models. For example, some models are fitted to large datasets composed of multiple isolates of each viral species [13,14]; even though only around 300 distinct viral species are known to infect humans [15,16]. Other models use a single reference sequence per species, leading to extremely small datasets [2,17]. While having more isolates can increase the sample size and has the potential to reveal more features, it also risks introducing near duplicates in the test set of data in the training set, thereby artificially increasing the performance metrics [18]. On the other hand, using only reference sequences produces smaller datasets that limit the modeling techniques available to improve performance, and for which the fitted models may miss key features of zoonotic emergence [19]. ML methods risk to overfit in either case: in large datasets because of over-representation of frequently sequenced viruses, and in small datasets because noise in the data is not averaged out. The disparate model evaluation metrics and data splitting protocols used by the community, summarized in Table 1, reflect different priorities and can vary significantly across datasets, especially in class-imbalanced settings. They can be difficult to compare directly [20–22]. Without

**Table 1. Heterogeneity of previously-described ML models that predict the likelihood of human infection based on viral genetic information. Our approach is shown in the bottom row, and cell colors are used to emphasize the similarities and differences between approaches. The ROC AUC of 0.73 on the holdout dataset was the best performance across all of our estimator types based on a reconstitution of the original training and testing sets produced at runtime by execution of the code at https://github.com/Nardus/zoonotic_rank at hash 42f15a07.**

| Model Type | Dataset (Source) | % Human Infecting | Metric Type | Metric Value | Data Split | Citation |
|---|---|---|---|---|---|---|
| k-nearest neighbors | 9,428 viral genomes (Virus-Host DB) | 13.1 | ROC AUC | 0.92 | cross-validated | [13] |
| deep neural network | 9,496 viral genomes | 13.7 | balanced accuracy | 91.7 | 80:10:10 train: test: validation | [24] |
| k-nearest neighbors | 9,496 viral genomes | 13.7 | balanced accuracy | 82.8 | 80:10:10 train: test: validation | [14] |
| support vector machine | 158 viral genomes (Virus-Host DB) | 50 | ROC AUC | 0.94 | 80:20 train: test split | [17] |
| gradient boosting | 861 viral genomes | 30.2 | ROC AUC | 0.77 (1), 0.73 (2) | (1) 70:15:15 train: calibration: test split over 100 iterations; (2) 0.73 on separate holdout dataset of 758 viruses (14.9% human infecting) | [2] |
| Random Forest, Extra Trees, gradient boosting, SVM | 849 viral genomes | 33.0 (human) 36.2 (primate) 73.5 (mammal) | ROC AUC | 0.784 ± 0.013 (human); 0.774 ± 0.015 (primate); 0.850 ± 0.020 (mammal) | holdout dataset of 736 viruses (19.0% human infecting; 27.6% primate infecting; 75.8% mammal infecting) | This Work |

consistency in datasets, data splits, and reporting strategies, direct comparisons between models cannot be made, or are misleading [19,23].

With this paper, we release an improved version of the dataset previously curated by others [4,25,26], and recently analyzed by Mollenzte *et al.* [2], by updating the assigned human infection potential and adding both potential mammal and primate infection labels. The extra host infection labels were added for a few reasons: 1) since we were already investing effort to check the primary literature for updated evidence of human infection, we often came across mammal and primate infection data incidentally; 2) there were previously published indications that broader taxonomic host ranks were more tractable for viral host determinations, especially in viruses infecting prokaryotes [17], and we wanted to investigate this more closely. The model performance metrics previously reported by [2] were based on repeated splits of a well-curated dataset of 861 records. In this work, we treat their entire curated dataset of 861 records as the starting point for refinement of the training dataset used here. Conversely, their holdout dataset, consisting of 758 records, was not curated but rather labelled based on the host organism from which a given virus was isolated. Here, we have hand curated these remaining 758 records based on the latest available literature, roughly doubling the number of available curated virus-host records available to the community. Before rebalancing the datasets, we also treated these 758 relabelled records as our original test dataset, rather than repeatedly splitting the original 861 curated records. We evaluated model performance on the improved dataset for common ML algorithms: random forest [27], extra-trees [27], gradient-boosted decision trees [28,29], and support vector machines [27] (Fig 4). We found the original data split had relatively poor performance (ROC AUC of 0.663 ± 0.070; 8 estimators, 10 random seeds), possibly due to imbalanced viral family representation, and demonstrate that rebalancing the training and testing sets improves performance (ROC AUC of 0.784 ± 0.013) when predicting human pathogens (Fig 1). Note that rebalancing the dataset reduces the phylogenetic distance between training and test sets, so improvements in estimator performance are likely the direct result of that. Furthermore, we see an expected increase in model performance when predicting mammal pathogens (ROC AUC of 0.850 ± 0.020; 8 estimators, 10 random seeds) relative to primate (ROC AUC of 0.774 ± 0.015) and human (ROC AUC of 0.784 ± 0.013) infecting pathogens. This improved dataset is a first step towards establishing a standard for determining human infectivity from genetic sequence data, and we suggest additional improvements that can be pursued towards this goal (see Discussion).

## 2 Results

### 2.1 Improvements to previous dataset

Careful review of the original datasets used in [2] (5th row in Table 1), and previously curated by several others [4,25,26], revealed some areas for potential improvement in the context of designing a dataset for longer term formal comparisons of models. For example, twelve accession numbers (unique identifiers) corresponding to four viruses were present in both the train and test datasets, which we removed from the test dataset but kept in the train dataset. Additionally, we removed

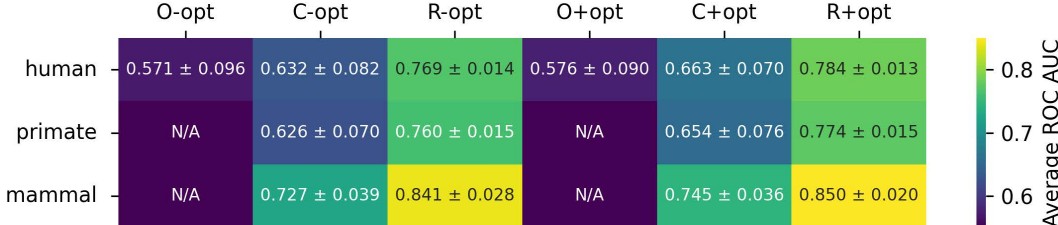

**Fig 1. Average estimator performances on test (ROC AUC across 10 seeds, with standard deviations) on different datasets in this work.** Estimators averaged include Random Forest, Extra Trees, gradient boosted trees, and support vector machines. The datasets include the original (O) from Mollentze, a corrected version (C) of the Mollentze dataset, and a rebalanced (R) version of the corrected dataset. Results are further divided by optimization of hyperparameters (+) or lack thereof (-).

viral genomes from either dataset if any of their accessions contained the keyword "partial" in their genbank file on NCBI, indicating an incomplete genome. Note, however, that most of these "partial" genomes are fairly complete, missing only some hard-to-sequence terminal nucleotides, and their removal is a fairly strict criterion. Finally, viral genomes that contained no coding sequences or only coding sequences not divisible by 3 were also removed, as amino acid based features could not be calculated for such coding sequences in [2] (see Discussion). The resulting corrected dataset now contains 849 (out of 861) genomes in the train set and 736 (out of 758) in the test set.

Since the holdout dataset used in [2] did not include review of the human infectivity of the viruses not isolated from humans, and because the knowledge of human infectivity of many viruses may have improved, we manually relabeled the viral dataset based on current literature (July 2025). While performing this review, we also determined primate and mammal infectivity for all viral species from the literature, as classifying against these targets may be useful for determining zoonotic emergence [30]. The updated abundance of human infecting viruses and the new abundances of primate and mammal-infecting viruses are summarized in Fig 2A and Table 1. In particular, we found 2.8% (train) and 4.1% (test) increases in human infectivity relative to the original work.

We also noticed that neither training nor test datasets had broad coverage over all viral families. This was a choice made by the authors as discussed in [2] and may account for their better performance observed on training than on the holdout set [2] (ROC AUC 0.77 vs 0.73, respectively). Although this may probe generalizability to a novel virus of unknown origin, a more likely scenario of zoonotic emergence would involve viruses in viral families known to contain species with the ability to infect humans or related vertebrates [31]. This suggests constraining training and testing sets to have similar representation across viral families. To achieve this, we reshuffled the train and holdout sets for each target host (human, primate, and mammal). The shuffle was unbiased with respect to viral family and preserved the total genomes and target label counts of the original datasets for comparison (Fig 2B). The additively smoothed ($\epsilon = 1 \times 10^{-9}$) Kullback-Leibler divergence (relative entropy) in viral family representation across training and test sets improved from 3.00 to 0.08 as a result of the shuffling, for human hosts. For primate and mammal host targets, the shuffling-based improvements in relative entropy were both 3.00 to 0.07.

### 2.2 Performance of ML models on dataset

Using our open source ML workflow available at https://github.com/lanl/ldrd_virus_work, we evaluated the following models on our dataset: random forest [27], extra trees [27], XGBoost [29], lightGBM (with both the boost and DART algorithms) [28,32], and support vector machines using the linear, polynomial, and radial basis function kernels [27]. Models were trained on the training set and their performance evaluated on the test set. It is notable that our ML workflow leverages many of the same genomic data features previously reported by Mollentze *et al.* [2], in addition to leveraging overlapping peptide kmers translated from the viral genomes. The following sections present two major findings on how data splits impact classification performance.

**2.2.1 Rebalancing dataset for viral family representation improves classification performance.** We observed that the models perform better (ROC AUC of 0.784 ± 0.013 vs. 0.663 ± 0.070; 8 ML models, 10 random seeds) on the task of predicting viral human infectivity on the rebalanced (shuffled) dataset and have more similar ROC curves (Figs 1, 3, 4) and that the corrected dataset yields a slightly higher average ROC AUC (0.663 ± 0.070) on the test set than the original dataset (0.576 ± 0.090) across all models. Note that improved estimator performance as a result of rebalancing the dataset is almost certainly a direct outcome of reducing the phylogenetic distance between training and test sets (relative entropy change from 3.00 to 0.08). Our XGBoost model underperforms the same estimator used by [2] on the original test set (0.657 ± 0.000; Fig 5, compared to 0.73) likely due to differences in our workflows (see Methods), one of which is the addition of peptide kmers in our array of features. Preliminary results with a single random seed produced from repository hash b305083c (https://github.com/lanl/ldrd_virus_work/pull/46) indicate that on the corrected data set that respects the original split used by Mollentze, prediction of human infection without hyperparameter optimization

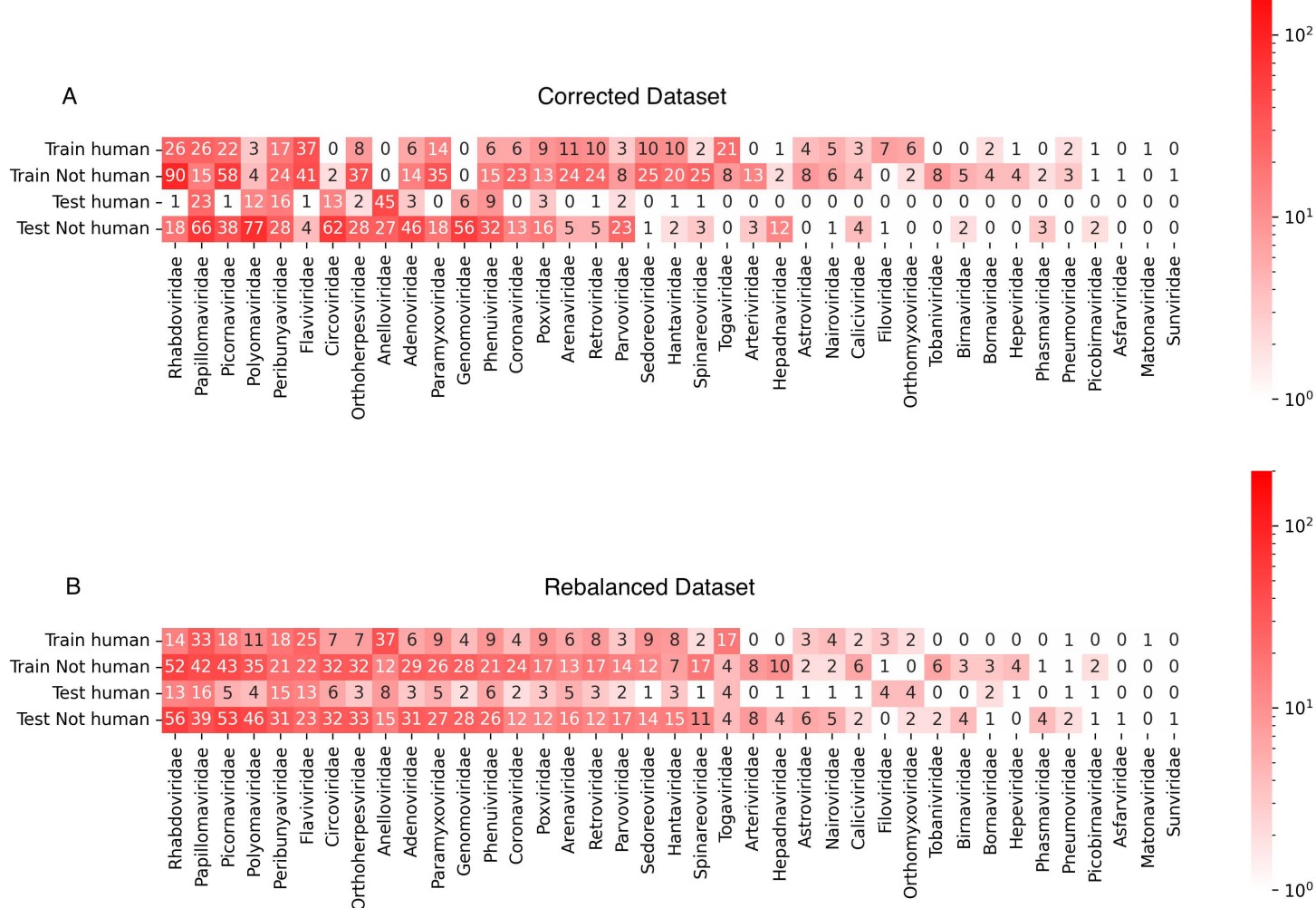

**Fig 2. Distribution of viral genomes in the datasets used in this work, categorized by human infectivity and training and test data split.** The datasets are improved versions of the original dataset analyzed by Mollentze et al. [2], and previously curated by several others [4,25,26], with specific improvements including removal of problematic genomes and updating known human infectivity **(A)**, or additionally rebalancing the datasets by random shuffling with preservation of human infectivity ratios **(B)**.

improves from an average ROC AUC of 0.63 ± 0.08 in the presence of kmer features (8060 top features retained) to 0.69 ± 0.06 with kmers stripped from the top features (285 features retained). Indeed, none of our estimators outperformed the original test set performance described by Mollentze *et al.* [2] (ROC AUC of 0.73), suggesting that the addition of peptide kmers to the array of features did not improve model performance on its own (Fig 5). In fact, some of the reported estimator performances on the original split were below random chance (AUC < 0.5). There have been previous reports of estimators performing below the random chance threshold on viral genetic classification problems, perhaps most notably on the binary task of determining if a given viral genetic segment is native or a recent acquisition via horizontal gene transfer [33]. In that work, the authors observed that some bacteriophage families consistently scored below random chance because their rates of horizontal gene transfer far exceeded those of other viral families in the trainig data, and so most of their native genetic segments actually appeared to be foreign. The generalization power of prediction across a

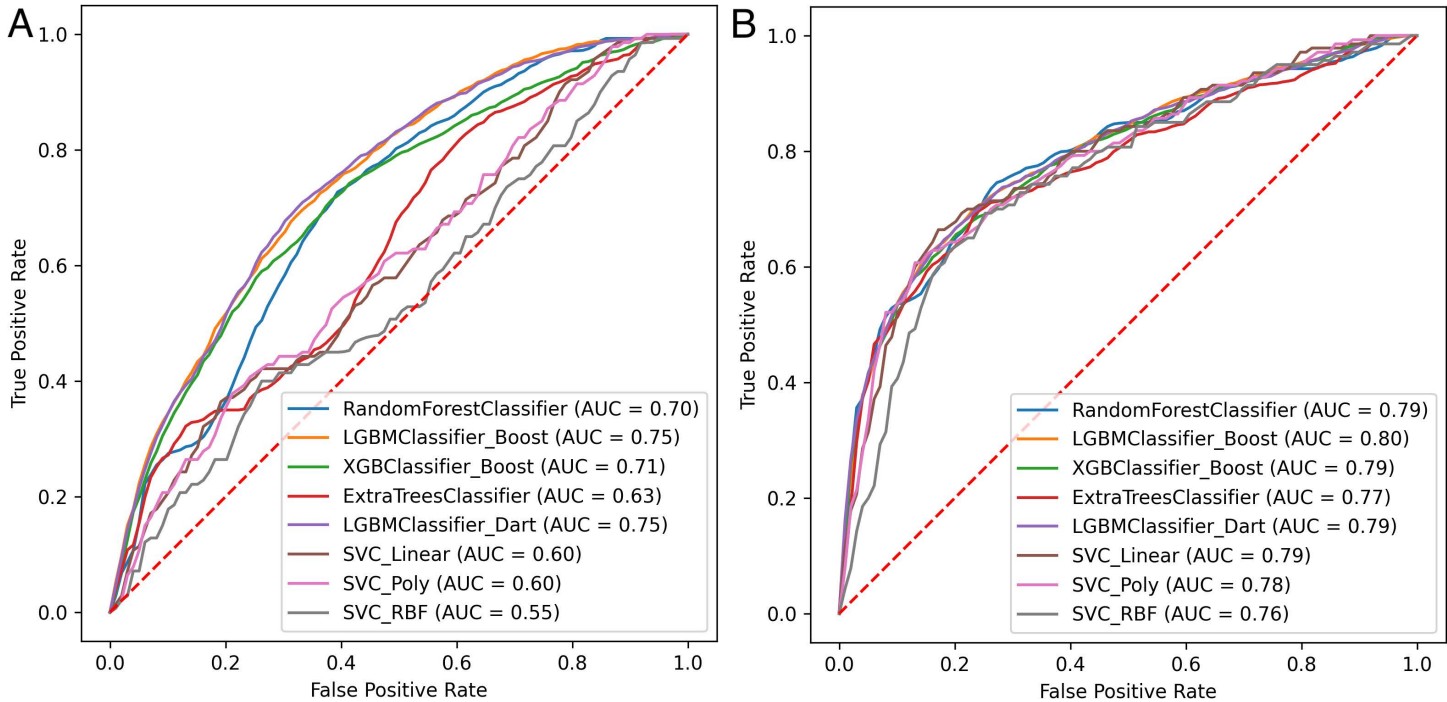

**Fig 3. The performance of common ML models was evaluated on the corrected (A) and rebalanced (B) datasets for prediction of viral human infectivity.** Features calculated using our workflow were similar to those used by Mollentze et al. [2], except that we additionally included peptide kmers. Hyperparameter optimization was performed for each model before training (see Methods). Mean ROC curves were calculated from prediction scores on the test data across 10 random seeds, with ROC values shown in the legends and standard deviations (SD) summarized in Fig 4. The dashed red line represents an estimator that is no better than random chance.

broad range of viral families may always be dubious given that viruses likely do not even share a common ancestor [10]. More broadly, it is fairly well established that when there are a small number of observations per class, weak signals, and high dimensionality (noise), data splits can yield performance below chance even under the null hypothesis because of anti-correlations between training and test sets [34]. Nonetheless, when combined with dataset rebalancing, our average estimator performance for prediction of viral human infectivity (0.784 ± 0.013) was greater than the originally published value (0.73). All evidence points to this difference being accounted for by the reduced phylogenetic distance between training and test sets in our rebalanced dataset. To decouple the possible influence of our additional kmer features, we performed some preliminary ablation studies (repository hash b305083c; https://github.com/lanl/ldrd_virus_work/pull/46). The eight estimators perform quite poorly when kmer features are included (7537 features used) on the original data split in the absence of hyperparameter optimization (single seed ROC AUC 0.571 ± 0.0795). Using the same data/feature set, but with kmers stripped out (257 features used), the performance improves to an average single seed ROC AUC of 0.668 ± 0.0435 across the eight estimators. Conversely, for the corrected and rebalanced data, the single seed averaged ROC AUC across the eight estimators was quite similar for the kmer-inclusive (0.766 ± 0.0141; 8916 features) and exclusive (0.762 ± 0.0367; 392 features) scenarios. It may be that kmer features are harmful to model performance on the original data split because of the larger number of out of sample viral family predictions and overfitting to kmers that are not broadly relevant across all viral families. A potential complication is that viruses, unlikely cellular life, likely do not share a common ancestor [10]. To investigate generalization power in a more rigorous way, we split the training and testing data such that there was no overlap in viral families between the two (repository hash 861ef2ff; relative entropy > 24)—performing six resampling trials with a random 60% of viral families represented in training, and the remaining 40% of

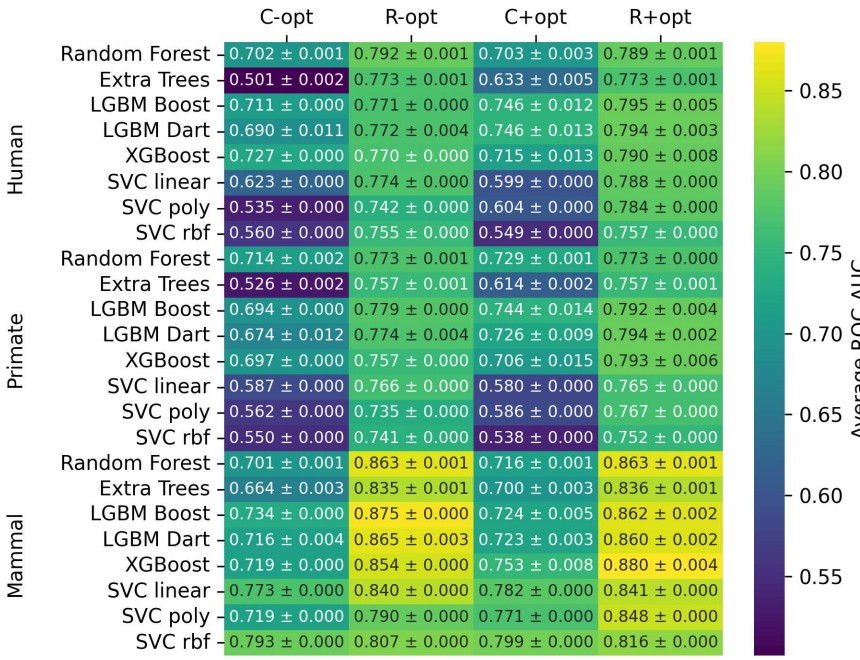

**Fig 4. Mean ROC AUC on test broken down by estimator, dataset (C=corrected; R=rebalanced), hyperparameter optimization status (-=non-optimized; += optimized), and host target.** Results are averages and standard deviations across ten random seeds.

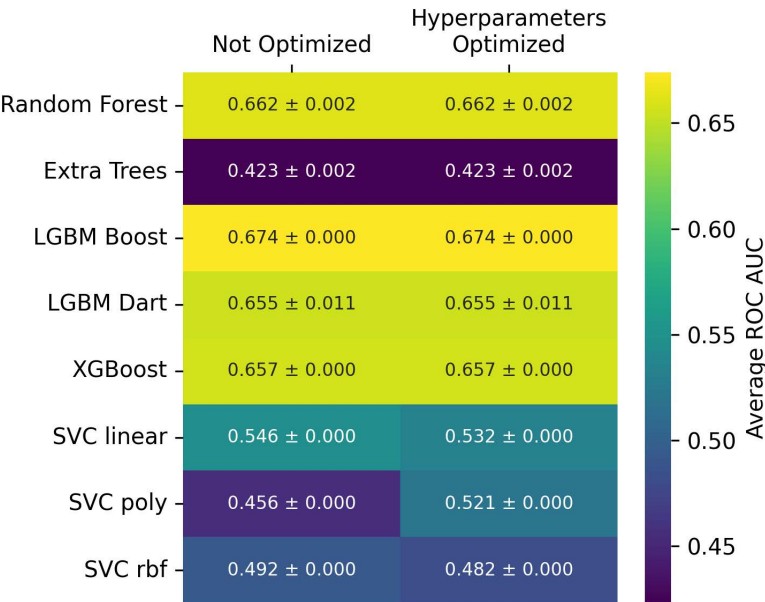

**Fig 5. Mean ROC AUC on test broken down by estimator and hyperparameter optimization status.** These results are for a dataset closely matched to the one originally leveraged by Mollentze *et al.*, which only has human target labels. Results are averages and standard deviations across ten random seeds.

viral families in test. The average ROC AUC for prediction of human infection on the test data was no better than random chance regardless of whether kmer features were present (0.50 ± 0.08) or not (0.50 ± 0.04).

**2.2.2  Changing Host Target Reveals Insights About Model Generalizability.**  To determine the ability of the model to generalize across broader host categories, we evaluated model performance for predicting primate and mammal infection (Fig 6). On the corrected dataset, the primate infection prediction achieved a mean ROC AUC of 0.654 ± 0.076. This performance is arguably indistinguishable from the results for the human target (0.663 ± 0.070) on the same dataset (Fig 1; 10 random seeds, 8 ML models). On the rebalanced dataset, the primate target achieved a mean ROC AUC of 0.774 ± 0.015, again arguably indistinguishable from the performance achieved for predicting human pathogens (0.784 ± 0.013). ML predictions of mammal infecting viruses achieve a mean ROC AUC of 0.745 ± 0.036 on the corrected dataset, which is arguably better than model predictions of human pathogens (0.663 ± 0.070) and primate pathogens (0.654 ± 0.076). On the rebalanced dataset, predictions for the mammal target achieved a mean ROC AUC of 0.850 ± 0.020, which is clearly higher than predictions for human (0.784 ± 0.013) and primate (0.774 ± 0.015) pathogens. Taken together, these results indicate that models trained on a balanced dataset have better performance and capture important features for a variety of host categories, but some of the general trends are still preserved even without shuffling, albeit with lower confidence.

## 3  Discussion

The need for classifiers to predict human infecting viruses from their genomic sequences alone is well motivated, yet progress towards their development has been slowed by inconsistencies in the datasets used, the metrics reported, and the data splitting practices [19,23] (Table 1). We showed that shuffling the viral genome datasets, to improve their balance of viral families between training and test data, tends to improve ML model performance for human infection prediction (ROC AUC of 0.784 ± 0.013 vs. 0.663 ± 0.070; 10 random seeds for 8 ML models). The datasets we share with the community here also include quality filtering sequences to avoid incomplete genomes, updating viral host infectivity to correct

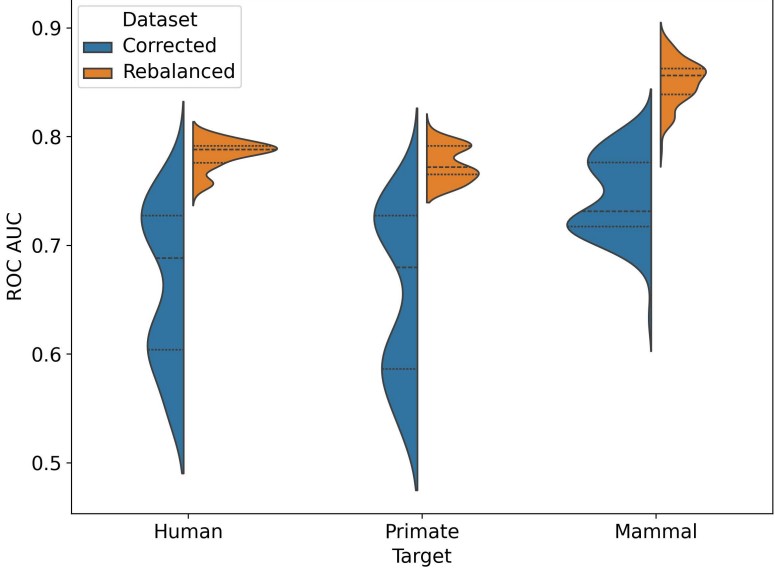

**Fig 6.  The performance of common machine learning models was evaluated on the corrected and rebalanced datasets for human, primate, and mammal host targets.** The ROC AUC for each model is reported for the predictions on test across 10 seeds. The violin plot displays the statistical distributions for ROC AUC across estimators, with dashed lines for the three quartiles–the 25th percentile, the median, and the 75th percentile.

erroneous data points based on the latest literature, and expanding to include primate and mammal host infection labels—infection of primates may ultimately be the phylogenetic barrier crossing of greatest interest. We used the datasets presented here to train multiple ML models, providing the community with baseline estimator performances via a code base and set of data that are open and reproducible. It remains an open question as to whether it would be preferable to probe ML model performance on datasets that are balanced on viral families or not—it depends, for example, on the interest in identifying truly novel pathogens relative to those that are quite similar to currently-known viruses.

We also found that models trained on mammalian infection show improved performance (ROC AUC 0.850 ± 0.020) over those trained on human (ROC AUC 0.784 ± 0.013) or primate (ROC AUC 0.774 ± 0.015) host labels. This may seem like a somewhat obvious phenomenon, given the broader range of the mammalian label, however we also observed a narrower spread of ROC-AUC of models trained on the corrected dataset as compared to human and primate hosts, which suggests that certain features of the mammalian host target may contribute to improved classification relative to non-mammal host infection. For both prokaryotic and eukaryotic virus host identification tasks, previously published results indeed suggest that machine learning model performance degrades at lower taxonomic ranks [17]. The authors specifically reported an average AUC of 0.86 ± 0.07 for bacterial host prediction at the phylum level, and only 0.67 ± 0.15 for host prediction at the species level. The authors noted that the larger dataset sizes at higher taxonomic ranks may contribute to improved predictive performance. In the future, it may be helpful to develop machine learning pipelines that are trained at different levels of taxonomy—for example, with an initial estimator filtering at the class (mammalia) level, another estimator at the order (primate) level, and a final estimator at the species (*H. sapiens*) level, each trained on their respective design matrices.

We recognize limitations of the work presented here which should be considered as these datasets develop. For example, we removed viruses if they did not contain any coding sequences divisible by 3 as these coding sequences are not used for feature calculation in [2]. However, viruses are known to abuse ribosomal frameshifting [35], and models with more robust feature calculation may be capable of making use of these genomes. Additionally, while we selected a train-test split for ease of comparison with results from [2], a smaller test set may be appropriate for other studies as we do not necessarily suggest a roughly even split between training and test data. That said, preliminary results (repository hash 22dcac1b; https://github.com/lanl/ldrd_virus_work/pull/45) when splitting the data 80% train: 20% test produced an average ROC AUC for prediction of human infection across the eight estimators of 0.779 ± 0.009 (single random seed) in the absence of hypeparameter optimization. This result is quite similar to the ROC AUC of 0.769 ± 0.014 we report across ten random seeds for the rebalanced human prediction data in the absence of hyperparameter optimization above. It may have been possible for us to achieve greater impact by demonstrating that our rebalanced datasets enable superior generalization power on a *third* dataset of recently-emerged viral pathogens of interest, for example avian influenza infection of dairy cows and humans [36]. This may also highlight the need for specialized challenge datasets where a small number of nucleotide changes in viral genomes is known to confer a change in host range, to probe the specific strengths and weaknesses of predictive models.

For prediction of prokaryotic viral hosts, iPHoP recently emerged as a promising tool that combines several different methods that span both host-based matching and phage-based matching [37]. It may be worth drawing some inspiration from their focus on matching host-based features, since our added virus-focused kmer features were not helpful to model performance in out of sample scenarios. However, direct usage of iPHoP on our data is not appropriate—the authors noted that not only is the tool explicitly expecting bacteriophages and archaeoviruses as input, but that when applied to 8128 eukaryotic virus genomes from RefSeq, iPHoP predicted a bacterial or archaeal host for 1018 viruses. Models trained on our eukaryotic virus-host datasets may suffer from the reverse false predictions, and it is clear that feature engineering in this domain still has a long way to go, even for distinguishing hosts that have very large phylogenetic separations. Specialized models focused on the identification of eukaryotic hosts, such as EvoMIL [38], occasionally claim exceptionally high performance in the identification of virus hosts. EvoMIL reports an ROC AUC for identification of human

infection of 0.93. The authors applied the ESM-1b protein language model to individual viral proteins to obtain sequence embeddings [39]. However, 1% of the ESM-1b training data includes viral proteins, and it may require substantial investigation to determine which virus-host test sets do not overlap with the protein language model training data to enable fair comparisons.

Although these results are a step towards establishing a benchmark dataset for prediction of viral host infectivity from genomic information, open questions remain. It has not yet been established if a larger dataset with multiple isolates per species is preferable to a smaller dataset with distinct representative samples. Our work assumes that using a smaller dataset is an acceptable tradeoff to mitigate potential information leakage that arises when multiple isolates per species are included into both the training and test datasets. Perhaps what our community needs most is a committee that curates and updates datasets used for assessing viral host infectivity, and a spirit of competition to drive model excellence. Examples of competitions include those hosted on Kaggle, and there are also several machine learning benchmark working groups and committees: ML Commons maintains machine learning performance benchmarks like MLPerf [40], and the common objects in context (COCO) consortium maintains a standard dataset for the computer vision community [41].

## 4 Methods

### 4.1 Dataset Improvement

The original dataset classes include 'Human-host' vs 'Not human-host' labels for viral infection [4,25,26] indicating whether a certain virus is known to infect a human or not. Initial results using these data classes show a large amount of false positives (Fig 7), which indicates that the true label may be 'human-host' but there may not be sufficient supporting data to make that claim. There is also potential for making phylogenetically informed classification decisions by relabeling the dataset to include classes that are aimed at different hosts. This modified target is based on the idea that whether a virus has spilled over into infecting primates may be a more informative indication of the zoonotic potential of the virus to infect humans. To this end, the viral dataset was relabeled from the original classes to include more broad categories using literature-based evidence. Each virus was relabeled as either "human," "Primate," "Mammal," "Avian," or "non-human" (i.e., insect, reptile, plant). Ground-truth relabeling

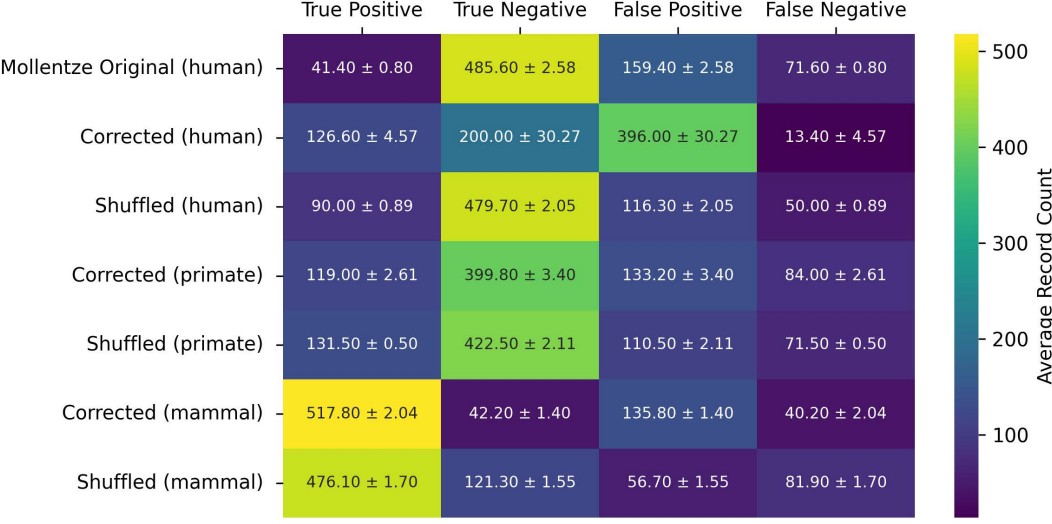

**Fig 7. Tabulation of confusion matrix data across hyperparameter optimized Random Forest estimators averaged across ten random seeds for their predictions on test set data.** The confusion matrix data for other estimators may be reproduced using the open source code accompanying this work. Average and standard deviation values are shown for different datasets. The threshold used for prediction was the equal error rate (EER).

decisions were made on a case-by-case basis by three experts using evidence of direct infection or presence of virus in host serum as positive confirmation, whereas evidence of infection in cell-culture was not considered sufficient. Additionally, mislabeled data-points were changed in certain cases based on new evidence. Note that while the 861 virus-host records used for reporting of performance metrics in [2] were previously curated, the 758 virus-host holdout records they used were not, so the careful verification and improvement of the 758 previously non-curated records was particularly important. The precise justifications for host infection statuses are described in the retarget.py module in our open source code, and in a human readable HTML table format at https://doi.org/10.6084/m9.figshare.31014793. In cases where the host label justifications are secondary sources, such as ICTV [42] or virushostdb [43], we did not find primary literature evidence to deviate from the host label provided by the secondary source. In some cases we deviated from the host labels assigned by the secondary sources, and typically cited the basis for these deviations in the primary literature, because they were not sufficient to meet our critera—in particular, cell culture propagation and the presence of antibodies in the host on their own were generally not considered sufficient. Many viruses are reported as propagated in the Vero cell line from primates, but there is otherwise no evidence of infection of primates and Vero cells lack many important host factors, including interferon [44]. We generally rejected the presence of antibodies in a single host on its own as sufficient evidence because exposure does not necessarily indicate viable viral replication in the host, as noted for example in one of our primary citations for an avian virus with antibodies found in human subjects [45]. However, note that we occasionally did allow antibody presence in the host as evidence when it was reported in a large sample of individuals of a species. Any evidence that the virus may be replicating in the host was generally accepted—we did not discern between specific methods of verifying the presence of viral nucleic acids or proteins. When the justification for a host label was specified as NaN, it was generally because the viral record was a duplicate or alias of another viral record that already had an assigned source of evidence for the host label. Of course, as new evidence of host infection statuses emerge, the dataset may be improved, and some of our records indeed have justification blocks that indicate the need for additional investigation. Beyond straightforward fixes to remove problematic or duplicated viral records, the latest literature evidence allowed us to relabel 21 viral species from non-infecting to human infecting, while no viral species were switched from human infecting to not. The final datasets used in the machine learning workflow are available in CSV files stored in project path viral_seq/data (at the DOI listed below), and supporting genomic data at another DOI also specified below.

### 4.2 ML Workflow

Our open source machine learning workflow (https://github.com/lanl/ldrd_virus_work) may be used to fully reproduce the results of this work, at commit hash `c21dfd8e6,` `git` release tag v0.1.0, and Zenodo DOI https://doi.org/10.5281/zenodo.17074080. Features were calculated for each viral genome in a manner that was strongly influenced by the original workflow published by *Mollentze et al.* [2]. We evaluated the datasets using the following models: `RandomForestClassifier,` `ExtraTreesClassifier,` and SVC with the linear, polynomial, and radial basis function kernels as implemented in the scikit-learn package [27], `LGBMClassifier` using both the default gradient boosting decision trees and DART [32] algorithms from the LightGBM package [28], and `XGBClassifier` from the XGBoost package [29]. Each model's hyperparameters were optimized for each dataset using the tree-structured Parzen estimator algorithm [46] for the `OptunaSearch` subpackage [47] in the Ray Tune [48] Python package. The number of steps of optimization was chosen for each model such that the performance of five independent seeds appeared to roughly plateau as measured by the mean ROC AUC using 5-fold cross-validation on training data stratified on the target (Fig 8). For `RandomForestClassifier,` `ExtraTreesClassifier,` and SVC kernels we used a broad range of values across many hyperparameters (Table 2). For `XGBClassifier` and `LGBMClassifier` we used the ranges suggested in the documentation of Amazon SageMaker [49,50], with the

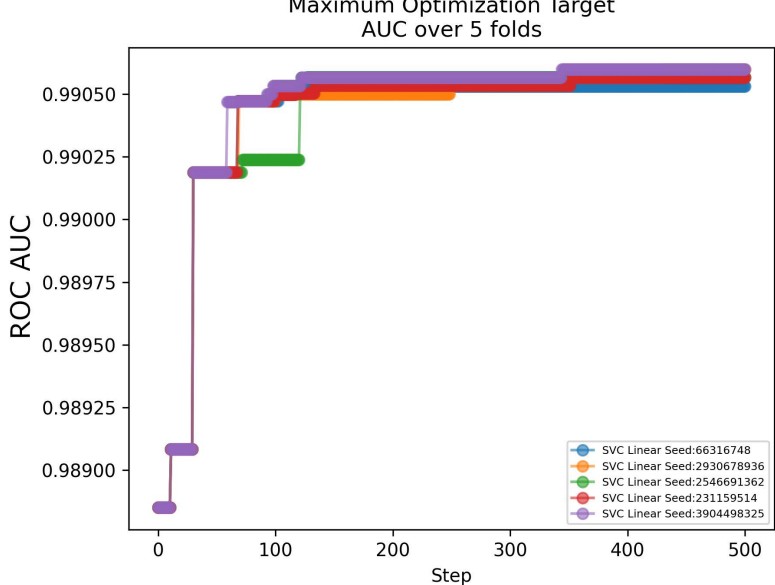

**Fig 8. Representative example of the process of determining the number of trials required for hyperparameter optimization.** The estimation is based on a plateau of performance improvement across five independent random seeds with 5-fold cross-validation mean ROC AUC as the metric. In this case, approximately 500 steps appears to be sufficient to optimize the hyperparameters for SVC with a linear kernel.

addition of the `drop_rate` and `skip_drop` hyperparameters over their full ranges for `LGBMClassifier` when using the DART algorithm. In addition to the software and libraries already mentioned, we also leveraged the following: SciPy [51], NumPy [52], polars [53], and pandas [54]. Plots were produced using matplotlib [55] and seaborn [56]. ROC AUC values were calculated as the average and standard deviation across the 8 estimators and 10 random seeds per condition.

## 5  Code availability

The ML code for this project, including the end-to-end workflow, is available on GitHub at https://github.com/lanl/ldrd_virus_work, and is CI-tested for CPython versions 3.10 through 3.13. The results of this work were specifically generated using the commit hash `c21dfd8`, release version v0.1.0 and DOI https://doi.org/10.5281/zenodo.17074080. The code was released under a GPL-3.0 license because it depends on the GPL-licensed `taxonomy-ranks` [57] project, which in turn depends on the GPL-licensed ete project [58]. We are open to relicensing our project to a more liberal scheme if these dependencies, which are used for phylogenetic heat map production, could be replaced.

## Acknowledgments

We thank Emma Goldberg and Anastasiia Kim for helpful feedback on the overall project design. This research used resources provided by the Darwin testbed at Los Alamos National Laboratory (LANL) which is funded by the Computational Systems and Software Environments subprogram of LANL's Advanced Simulation and Computing program (NNSA/DOE). This research used resources provided by the Los Alamos National Laboratory Institutional Computing Program, which is supported by the U.S. Department of Energy National Nuclear Security Administration under Contract 89233218CNA000001.

**Table 2. The range of hyperparameter values over which optimizations were performed.** $n$ represents the number of records in the design matrix, while $f$ represents the number of features in the design matrix. In many cases the samples were drawn over the intervals using a uniform distribution, but other approaches were also used—see the open source machine learning workflow for details.

| Hyperparameter | Random Forest | Extra Trees | LGBM Boost | LGBM DART | XGBoost | SVC linear | SVC poly | SVC rbf |
|---|---|---|---|---|---|---|---|---|
| max_samples | $[1/n, 1.0)$ | $[1/n, 1.0)$ | N/A | N/A | N/A | N/A | N/A | N/A |
| min_samples_leaf | $[1/n, \min(1, 10/n))$ | $[1/n, \min(1, 10/n))$ | N/A | N/A | N/A | N/A | N/A | N/A |
| min_samples_split | $[1/n, \min(1, 300/n))$ | $[1/n, \min(1, 300/n))$ | N/A | N/A | N/A | N/A | N/A | N/A |
| max_features | $[1/f, \min(1, 2\sqrt{f}/f))$ | $[1/f, \min(1, 2\sqrt{f}/f))$ | N/A | N/A | N/A | N/A | N/A | N/A |
| criterion | gini \| log_loss | gini \| log_loss | N/A | N/A | N/A | N/A | N/A | N/A |
| class weight | None \| balanced \| balanced_subsample | None \| balanced \| balanced_subsample | N/A | N/A | N/A | None \| balanced | None \| balanced | None \| balanced |
| max_depth | $\infty\|[1, 30]$ | $\infty\|[1, 30]$ | $[15, 100)$ | $[15, 100)$ | N/A | N/A | N/A | N/A |
| num_leaves | N/A | N/A | $[10, 100)$ | $[10, 100)$ | N/A | N/A | N/A | N/A |
| learning_rate | N/A | N/A | $[0.001, 0.01)$ | $[0.001, 0.01)$ | $[0.1, 0.5)$ | N/A | N/A | N/A |
| subsample | N/A | N/A | $[0.1, 1.0)$ | $[0.1, 1.0)$ | $[0.5, 1.0)$ | N/A | N/A | N/A |
| subsample_freq | N/A | N/A | $[0, 10)$ | $[0, 10)$ | N/A | N/A | N/A | N/A |
| min_child_samples | N/A | N/A | $[10, 200)$ | $[10, 200)$ | N/A | N/A | N/A | N/A |
| colsample_bytree | Data | Data | $[0.1, 1.0)$ | $[0.1, 1.0)$ | Data | Data | Data | Data |
| reg_alpha | N/A | N/A | N/A | N/A | $[0.001, 1000)$ | N/A | N/A | N/A |
| min_child_weight | N/A | N/A | N/A | N/A | $[0.001, 120)$ | N/A | N/A | N/A |
| n_estimators | constant | constant | constant | constant | $[1, 4000)$ | constant | constant | constant |
| drop_rate | N/A | N/A | N/A | $[0.0, 1.0)$ | N/A | N/A | N/A | N/A |
| skip_drop | N/A | N/A | N/A | $[0.0, 1.0)$ | N/A | N/A | N/A | N/A |
| C | N/A | N/A | N/A | N/A | N/A | $[10^{-4}, 10^4)$ | $[10^{-4}, 10^4)$ | $[10^{-4}, 10^4)$ |
| degree | N/A | N/A | N/A | N/A | N/A | N/A | $[2, 6)$ | N/A |
| gamma | N/A | N/A | N/A | N/A | N/A | N/A | $[10^{-4}, 100)$ | $[10^{-4}, 100)$ |
| coef0 | N/A | N/A | N/A | N/A | N/A | N/A | $[0.0, 10.0)$ | N/A |

## Author contributions

**Conceptualization:** Tyler Reddy, Nick Hengartner.

**Data curation:** Tyler Reddy, Aaron R Hall, Adam Witmer.

**Formal analysis:** Tyler Reddy, Nick Hengartner.

**Funding acquisition:** Tyler Reddy, Nick Hengartner.

**Investigation:** Tyler Reddy, Aaron R Hall, Adam Witmer, Nick Hengartner.

**Methodology:** Tyler Reddy, Aaron R Hall, Adam Witmer, Nick Hengartner.

**Project administration:** Tyler Reddy, Nick Hengartner.

**Resources:** Tyler Reddy.

**Software:** Tyler Reddy, Austin Schneider, Aaron R Hall, Adam Witmer.

**Supervision:** Tyler Reddy, Nick Hengartner.

**Validation:** Tyler Reddy, Nick Hengartner.

**Visualization:** Tyler Reddy.

**Writing – original draft:** Tyler Reddy, Nick Hengartner.

**Writing – review & editing:** Tyler Reddy, Austin Schneider, Aaron R Hall, Adam Witmer, Nick Hengartner.

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
