## [Decision Letter · Decision Letter 0]

26 Nov 2025

PCOMPBIOL-D-25-01998

An Improved Dataset for Predicting Mammal Infecting Viruses from Genetic Sequence Information

PLOS Computational Biology

Dear Dr. Reddy,

Thank you for submitting your manuscript to PLOS Computational Biology. After careful consideration, we feel that it has merit but does not meet PLOS Computational Biology's publication criteria as it currently stands. Therefore, we invite you to submit a revised version of the manuscript that addresses the points raised during the review process.

Briefly, both reviewers thought the manuscript makes a valuable contribution to the field, but both identified substantial issues they wish addressed in terms of statistical methodology, comparisons to state of the art, and dataset provenance and description.  Based on the major changes requested, if you need more time to revise please just let us know.

We look forward to receiving your revised manuscript.

Kind regards,

Peter M Kasson

Academic Editor

PLOS Computational Biology

Shaun Mahony

Section Editor

PLOS Computational Biology

**Journal Requirements:**

At this stage, the following Authors/Authors require contributions: Aaron R Hall, Adam Witmer, Nick Hengartner, and Austin Schneider. Please ensure that the full contributions of each author are acknowledged in the "Add/Edit/Remove Authors" section of our submission form.

**Reviewers' comments:**

Reviewer's Responses to Questions

**Comments to the Authors:**

**Please note that one review is uploaded as an attachment.**

Reviewer #1: This is an interesting manuscript and represents a lot of work (especially on data curation), but is too brief in key places. More detail about the datasets used, and the curation performed (see below) would help highlight the important contributions of this manuscript.

I do not think our publication (Mollentze et al, 2021, PLOS Biology) should be cited as the source of the data used here – we published a model based on available data, and did not do any curation ourselves apart from picking a representative genome for each entry. Instead, please give credit to the original authors as described in our publication (primarily Olival et al. and Woolhouse & Brierley, with a very small number of updates to these in Mollentze et al. 2020). Note, however, that this applies to the curated data only (N = 861 viruses).

On a related note, I was more than half way through the manuscript before the link between the “training” and “testing” datasets mentioned here and any of the data we used in Mollentze et al. became clear (and then only by comparing dataset sizes) – I think for general readers this task will be impossible currently. We had 1000 randomly split datasets called “test” and “train”, and a completely independent set of viruses which were treated as an additional ad hoc “holdout” dataset – it appears this is what is being referred to as our “testing” data here, even though none of our model performance metrics were based on these data. Given the importance of dataset splits uncovered in this manuscript, I think it is important to provide readers with more context – these were not formal data splits as is being implied here, but two completely independent datasets from different sources, as explained in Mollentze et al. Only one of these datasets had high quality labels suitable for model training and testing (an amalgamation of data curated in various other studies, as described above; the other dataset being all remaining viruses), so adding this additional context would also help highlight that by curating the data in the “holdout” dataset, the authors have nearly doubled the amount of available data - a pretty significant change for the entire field.

Other comments

Abstract:

It is unclear what is meant by models having a different “baseline” in this context.

It remains unclear from the abstract whether the improved performance is from the data improvements, a different data splitting scheme, or a different model (since 8 models are mentioned).

I would suggest an additional sentence explaining why the broader host labels might be useful (and any results suggesting that they are). I was also missing a general conclusion sentence.

Page 1:

First paragraph: We can predict many (most?) virus protein structures from the sequence already, so this is no longer a good example of a virus feature that is hard to obtain. I would agree that there are many phenotypic measures which do fit this description however.

Final paragraph: More detail is needed here - what are these new labels for / what is the reasoning behind adding them? This is currently only explained in the discussion.

Page 3:

Is this accuracy measure for the Mollentze et al. holdout data based on curation of these data done in the present work? If so that should be made clear. In the original manuscript, we quoted the number of viruses first detected in humans that were correctly predicted as human-infecting (70.8%), but noted that many of the other viruses likely are capable of human infection too, even if first detected in another host.

Page 4:

In addition to the need for more context about these datasets mentioned above, note that simply using a keyword search for “partial” is a very strict filter for full genomes. At least in the curated part of our data, genomes that still contain that keyword are “near complete”, with all required ORFs and nearly the right length – generally what is missing is simply the hard-to-sequence terminals. I’d be surprised if this filter makes much of a difference (for the curated data), but agree that for a formal dataset intended for future comparisons it is simpler to remove such sequences. However, I would not consider their inclusion in Mollentze et al. an “inaccuracy”.

Page 11:

First paragraph: If it is true that features inherent to mammalian hosts make their viruses easy to distinguish from non mammal-infecting viruses, this would suggest a hierarchical approach would be even more accurate - first predict which viruses can infect mammals, then predict which of those can infect primates/humans. This seems like a future direction worth discussing.

Second paragraph: Given that the previous division of datasets was entirely arbitrary, I’m not convinced there needs to be any effort to maintain these sizes when re-balancing. To me it would be far more interesting to see how much models improved when using the large amount of additional data to it’s fullest extent, to have a model based on much more training data.

Page 12:

The first part of this page does a good job of making the case for the less resolved host labels – I would move this much earlier in the manuscript to make this reasoning clearer from the start.

As the dataset is repeatedly highligted as the main point of this manuscript, much more information on the curation process and evidence standards considered will be needed. For example, what methods of confirmation of virus presence were considered sufficient (e.g. sequencing only, PCR- based, etc.)? Further, what source(s) of evidence were used? All I could find was the comments in retarget.py, but comments in a python script is not a very accessible format. Based on the comments on that script however, I am concerned – there seems to be extensive use of secondary sources like ICTV reports or virushostdb. Were the ultimate sources of such entries checked for accuracy?

Page 15:

Data availability – I could not find a list of citations to support the labels added here. Please provide these in a human-readable format to allow verification and extension by others.

Reviewer #2: See attachment

**Have the authors made all data and (if applicable) computational code underlying the findings in their manuscript fully available?**

The PLOS Data policy requires authors to make all data and code underlying the findings described in their manuscript fully available without restriction, with rare exception (please refer to the Data Availability Statement in the manuscript PDF file). The data and code should be provided as part of the manuscript or its supporting information, or deposited to a public repository. For example, in addition to summary statistics, the data points behind means, medians and variance measures should be available. If there are restrictions on publicly sharing data or code —e.g. participant privacy or use of data from a third party—those must be specified.requires authors to make all data and code underlying the findings described in their manuscript fully available without restriction, with rare exception (please refer to the Data Availability Statement in the manuscript PDF file). The data and code should be provided as part of the manuscript or its supporting information, or deposited to a public repository. For example, in addition to summary statistics, the data points behind means, medians and variance measures should be available. If there are restrictions on publicly sharing data or code —e.g. participant privacy or use of data from a third party—those must be specified.requires authors to make all data and code underlying the findings described in their manuscript fully available without restriction, with rare exception (please refer to the Data Availability Statement in the manuscript PDF file). The data and code should be provided as part of the manuscript or its supporting information, or deposited to a public repository. For example, in addition to summary statistics, the data points behind means, medians and variance measures should be available. If there are restrictions on publicly sharing data or code —e.g. participant privacy or use of data from a third party—those must be specified.requires authors to make all data and code underlying the findings described in their manuscript fully available without restriction, with rare exception (please refer to the Data Availability Statement in the manuscript PDF file). The data and code should be provided as part of the manuscript or its supporting information, or deposited to a public repository. For example, in addition to summary statistics, the data points behind means, medians and variance measures should be available. If there are restrictions on publicly sharing data or code —e.g. participant privacy or use of data from a third party—those must be specified.

Reviewer #1: Yes

Reviewer #2: Yes

PLOS authors have the option to publish the peer review history of their article (what does this mean?). If published, this will include your full peer review and any attached files.). If published, this will include your full peer review and any attached files.). If published, this will include your full peer review and any attached files.). If published, this will include your full peer review and any attached files.

...

Reviewer #1: No

Reviewer #2: No

**Figure resubmission:**

**Reproducibility:**



---

## [Decision Letter · Decision Letter 1]

2 Mar 2026

PCOMPBIOL-D-25-01998R1

An Improved Dataset for Predicting Mammal Infecting Viruses from Genetic Sequence Information

PLOS Computational Biology

Dear Dr. Reddy,

Thank you for submitting your manuscript to PLOS Computational Biology. Both reviewers thought that the majority of their comments were well addressed.  One reviewer has an additional suggestion to improve the robustness of the conclusions.  I would recommend addressing this recommendation; if the authors believe it would require an infeasible amount of computational effort, please justify that and and concomitantly soften the conclusions relating to that analysis.  If the more robust analysis is feasible, though, it would lead to a stronger paper.  We look forward to receiving your response and moving this manuscript forward--whichever path you choose, it's pretty close.

We look forward to receiving your revised manuscript.

Kind regards,

Peter M Kasson

Academic Editor

PLOS Computational Biology

Shaun Mahony

Section Editor

PLOS Computational Biology

**Reviewers' comments:**

Reviewer's Responses to Questions

**Comments to the Authors:**

Reviewer #1: The authors have addressed all of my previous comments.

Reviewer #2: Thank you for the revised manuscript. The revision addresses many of my previous concerns,

and the paper is substantially improved. In particular, the authors now clarify that the

performance gain on the rebalanced split is likely driven by reduced phylogenetic distance

(rather than a stronger model), remove earlier significance claims based on the sign test, add

preliminary k-mer ablation analysis, and improve data transparency by providing clearer

curation criteria and a human-readable label/citation table.

I have the remaining suggestion that could further strengthen the paper:

• Strengthen the ablation evidence with multi-seed or repeated resampling runs. The new kmer

ablation in Section 2.2.1 is helpful, but it is currently presented as preliminary and singleseed.

Because this result is central to the interpretation (k-mers appearing harmful on the

harder split), please repeat the same ablation across multiple random seeds and, if feasible,

repeated resampling/split repeats. Reporting averaged results (with uncertainty) would

make this conclusion much more robust.

**Have the authors made all data and (if applicable) computational code underlying the findings in their manuscript fully available?**

The PLOS Data policy requires authors to make all data and code underlying the findings described in their manuscript fully available without restriction, with rare exception (please refer to the Data Availability Statement in the manuscript PDF file). The data and code should be provided as part of the manuscript or its supporting information, or deposited to a public repository. For example, in addition to summary statistics, the data points behind means, medians and variance measures should be available. If there are restrictions on publicly sharing data or code —e.g. participant privacy or use of data from a third party—those must be specified.requires authors to make all data and code underlying the findings described in their manuscript fully available without restriction, with rare exception (please refer to the Data Availability Statement in the manuscript PDF file). The data and code should be provided as part of the manuscript or its supporting information, or deposited to a public repository. For example, in addition to summary statistics, the data points behind means, medians and variance measures should be available. If there are restrictions on publicly sharing data or code —e.g. participant privacy or use of data from a third party—those must be specified.requires authors to make all data and code underlying the findings described in their manuscript fully available without restriction, with rare exception (please refer to the Data Availability Statement in the manuscript PDF file). The data and code should be provided as part of the manuscript or its supporting information, or deposited to a public repository. For example, in addition to summary statistics, the data points behind means, medians and variance measures should be available. If there are restrictions on publicly sharing data or code —e.g. participant privacy or use of data from a third party—those must be specified.requires authors to make all data and code underlying the findings described in their manuscript fully available without restriction, with rare exception (please refer to the Data Availability Statement in the manuscript PDF file). The data and code should be provided as part of the manuscript or its supporting information, or deposited to a public repository. For example, in addition to summary statistics, the data points behind means, medians and variance measures should be available. If there are restrictions on publicly sharing data or code —e.g. participant privacy or use of data from a third party—those must be specified.

Reviewer #1: Yes

Reviewer #2: Yes

PLOS authors have the option to publish the peer review history of their article (what does this mean?). If published, this will include your full peer review and any attached files.). If published, this will include your full peer review and any attached files.). If published, this will include your full peer review and any attached files.). If published, this will include your full peer review and any attached files.

...

Reviewer #1: No

Reviewer #2: No

**Figure resubmission:**
---

## [Editor Report · Decision Letter 2]

15 Mar 2026

Dear Dr. Reddy,

We are pleased to inform you that your manuscript 'An Improved Dataset for Predicting Mammal Infecting Viruses from Genetic Sequence Information' has been provisionally accepted for publication in PLOS Computational Biology.  Thanks so much for the thoughtful response to the final reviewer query.

Best regards,

Peter M Kasson

Academic Editor

PLOS Computational Biology

Shaun Mahony

Section Editor

PLOS Computational Biology

---

## [Editor Report · Acceptance letter]

PCOMPBIOL-D-25-01998R2

An Improved Dataset for Predicting Mammal Infecting Viruses from Genetic Sequence Information

Dear Dr Reddy,

I am pleased to inform you that your manuscript has been formally accepted for publication in PLOS Computational Biology. Your manuscript is now with our production department and you will be notified of the publication date in due course.

With kind regards,

Anita Estes
